# Towards Sustainable Cities: Utilizing Computer Vision and AI for Efficient Public Lighting and Energy Management

**Anderson Silva Vanin** [1] and **Peterson Belan** [1,*]

Informatics and Knowledge Management Post Graduate Program, Nove de Julho University,
São Paulo 01525-000, Brazil; profandersonvanin@gmail.com
* Correspondence: pbelan@gmail.com; Tel.: +55-11-976336694

**Abstract:** This study showcases the optimization of public lighting systems using computer vision with an emphasis on the YOLO algorithm for pedestrian detection, aiming to reduce energy expenses. In a time when the demand for electricity is escalating due to factors like taxes and urban expansion, it is imperative to explore strategies to cut costs. One pivotal area is public lighting management. Presently, governments are transitioning from sodium vapor lighting to LED lamps, which already contributes to decreasing consumption. In this scenario, computer vision systems, particularly using YOLO, have the potential to further reduce consumption by adjusting the power of LED lamps based on pedestrian traffic. Additionally, this paper employs fuzzy logic to calculate lamp power based on detected pedestrians and ambient lighting, ensuring compliance with the NBR 5101:2018 standard. Tests with public surveillance camera images and simulations validated the proposal. Upon implementing this project in practice, a 45% reduction in public lighting consumption was observed compared to conventional LED lighting.

**Keywords:** pedestrian detection; YOLO (you only look once); fuzzy logic; electric energy management; public lighting control





## 1. Introduction

Within the state of São Paulo, one of the main categories of electricity consumption is public lighting, accounting for 2.4% of the total. According to the Energy Research Company [1], this percentage equates to about 3.2 thousand GWh of the energy consumption of 132 thousand GWh in the state. In the context of smart cities, public lighting should meet quality standards that ensure the safety of vehicles and pedestrian traffic at night, in addition to providing an adequate environment during the lack of natural light. These needs demand the proposal of new solutions to avoid the waste of electricity [2,3].

Replacing incandescent bulbs with LEDs can result in considerable energy savings, durability, efficiency, and sustainability, reducing energy consumption by approximately 30% [1]. However, this strategy does not take into account that lights often remain on at full power for long periods, generating unnecessary costs. This happens because they use obsolete switch systems that, based on regulatory standard NBR 5101:2018 [4], simply turn the system on or off depending on the presence of natural light.

Currently, the management of the operation of these systems is conducted through light photosensors that turn the lamps on or off according to the incidence of natural light, always at 100% of their power. However, they cannot determine the presence or the number of people on a public road, leaving room for more effective control through the use of computer vision techniques [5].

The use of intelligent techniques, such as computer vision, neural networks, and power controllers, in conjunction with system optimization can lead to more efficient management and cost reduction [3,6,7]. In this regard, the work proposed by Yigitcanlar et al. [3] presents a literature review highlighting the benefits and risks of using artificial intelligence

in the evolution of smart cities. Meanwhile, in the study proposed by Mohandas [7], a model for lighting control using only motion sensors with fuzzy logic was presented. Lastly, in the work of Islan et al. [6], an IoT-based system was proposed to track traffic accidents and notify the relevant authorities, sending the location and the car's number plate. Computer vision uses techniques that aim to estimate or explain the geometric and dynamic properties of an object from digital images [8]. Numerous applications for people recognition have been described in the literature, which use a variety of techniques [9,10]. The implementation of such techniques is part of the concept of smart cities [11], which seek to improve the efficiency of their operations and services, connecting with their citizens.

Improving energy control and management systems is also aligned with the Sustainable Development Goals (SDGs) of the 2030 Agenda, approved by UN members in September 2015. Goal 7, "Affordable and Clean Energy", is directly related to energy efficiency and universal access to reliable and affordable energy services [12].

Therefore, this work proposes the development of an effective methodology for optimizing the public lighting system, using computer vision and artificial intelligence techniques. This includes the analysis of pedestrian demand on roads and the intensity of natural light with convolutional neural networks, and the use of fuzzy logic for the dimming of the power of public lighting fixtures. The implementation of this methodology is expected to result in significant electricity savings for public lighting.

### 1.1. Literature Review

Significant control over electricity savings can be achieved by adjusting the brightness power of lamps based on the presence of pedestrians. For this, computational models are used to manage and control these devices [13]. In the experiment conducted by Galindo et al. [13], the presence or absence of humans was the only factor considered. When there was a presence of pedestrians, the lighting was adjusted to 100% brightness and, in the absence of pedestrians, reduced to 20%. This approach has similar functionality to presence sensors. Many computer vision techniques, combined with Convolutional Neural Networks, have been proposed to detect pedestrians on public roads. However, these techniques often encounter difficulties in identifying objects that are very close to each other in the same image [13].

The YoloV3 framework has proven to be effective in identifying small and close objects in an image, as observed in the literature review [14]. YoloV3 is an artificial neural network that uses deep learning and has been used in several pieces of research as an alternative for the accurate and fast detection of people and animals.

The article presented by Ahmed et al. [15] proposed a model to solve the problem of detecting small objects in surveillance camera monitoring systems using the YOLO framework. A comparison was made to evaluate the error rate in correct pedestrian detection, using a database of images called INRIA. Various types of detection algorithms, such as HOG, ConvNet, and YOLO, were employed. The experiments indicated that the system needs to be trained with large datasets to minimize false detections and the error rate, which are crucial for the correct detection of the number of pedestrians and animals.

Mohandas et al. [7] proposed a model that uses a lighting sensor, a motion sensor, and a PIR sensor as inputs. This model includes an artificial neural network and a Fuzzy logic controller, which assists in making decisions about energy use based on demand (light and presence). However, this model does not address the use of computer vision to quantify pedestrians.

Based on the bibliographic research, the works of Dizon et al. [16] and Mohandas et al. [7] are the closest to our study. However, they use three types of sensors that are activated by the movement of pedestrians and by the condition of natural light, but do not take into account the amount of pedestrians. They alternate the power of the lighting units between 20% and 100%, functioning almost like an on/off switch.

Islam et al. [6] proposed an innovative and economical system based on IoT to track accidents from the authorities' control room. The system can detect traffic accidents and

notify the competent authority by sending the location and the car number. In addition, an emergency button with face detection has been incorporated to help anyone in danger. The self-powered system can remotely monitor the streetlight and increase or decrease its intensity. Deep learning was used for implementation; the system can be operated by a mobile application from anywhere at any time, and the data will be periodically updated on the server.

Akindipe et al. [17] developed a holistic framework for intelligent road lighting in small cities, using a case study in Laramie, Wyoming, USA. The framework encompasses technical analysis, economic evaluation, and public survey. In total, 21 lighting units in a staggered arrangement are suggested, with two energy options: grid-only or a hybrid system with solar energy and grid electricity for charging electric vehicles. The analysis reveals the need for public-private partnerships to finance such projects. Although the grid-only option is more economically feasible, residents prefer the hybrid system.

*1.2. Public Lighting Cost Management*

Cost management in public lighting faces the challenge of efficiently using lighting technologies, since two-thirds of these systems still rely on obsolete technologies, consuming more energy than necessary [18]. Traditional energy-saving techniques, such as partially or completely turning off streetlights, have a negative impact on lighting uniformity and lamp lifespan. One solution to reduce consumption is to control the on–off time interval of lighting, which can be managed using photocell relays or astronomical time relays [18].

Emerging technologies provide significant advances in energy management. [13] presented a system that, through computer vision techniques, alternates lighting power based on the detection of humans. Mohandas et al. [7] proposed a model that employs sensors and an Artificial Neural Network coupled with a fuzzy controller; however, their system does not adjust the light based on the number of pedestrians.

New research is exploring energy savings in public lighting using hybrid modeling systems and IoT technologies for brightness dimming based on the presence of pedestrians [6,17,19]. Dizon et al. [16], for example, proposed a model where illuminance progressively decreases with the distance from the pedestrian, saving energy and improving user experience, as shown in Figure 1. These innovative approaches illustrate the potential to enhance public lighting cost management.

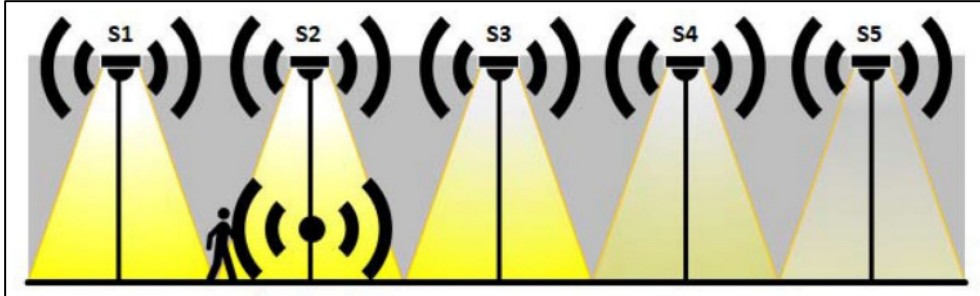

**Figure 1.** Adaptive lighting scheme for pedestrians [16]. S1–S5 Represents the region of each post, where each one has a sensor, separating the lighting zones.

However, it is important to note that public lighting follows a standardization rule, specifically the ABNT NBR 5101:2018 [4], which is applicable in our region. This regulation classifies the volume of pedestrians present on the roads into four groups: no traffic, light, medium, and intense, considering the type of road according to Table 1.

**Table 1.** Pedestrian traffic on roads [4].

| Classification | Pedestrians Crossing Roads with Motorized Traffic |
| --- | --- |
| No traffic (S) | Arterial roads |
| Light (L) | Average residential roads |
| Medium (M) | Secondary commercial roads |
| Intense (I) | Main commercial roads |

The same rule, considering the type of road, provides a classification for the minimum luminance necessary for pedestrians, as indicated in Table 2.

**Table 2.** Minimum average illuminance for each lighting class [4].

| Lighting Class | Minimum Average Illuminance (lux) |
| --- | --- |
| P1 | 20 |
| P2 | 10 |
| P3 | 5 |
| P4 | 3 |

The categories from P1 to P4 refer to the type of road to be considered based on pedestrian flow. Therefore, for a specific type of road, the minimum average amount of illuminance can be adjusted according to the number of pedestrians.

## 2. Materials and Methods

### 2.1. Framework YOLO

YOLO, which stands for "You Only Look Once", is a framework known for its speed and accuracy in real-time object detection. The basic premise of YOLO is that image processing occurs only once; that is, a single neural network performs all the work of predicting the detected objects [14,20].

Unlike traditional object detection methods, YOLO, as a unified convolutional network, simultaneously predicts multiple "bounding boxes" and class probabilities for these boxes. Training occurs on complete images, directly optimizing detection performance. Even when parts of the object are occluded, YOLO is able to identify the object through a direct regression from the image pixels to the bounding box coordinates, as well as their original dimensions and location, as shown in Figure 2.

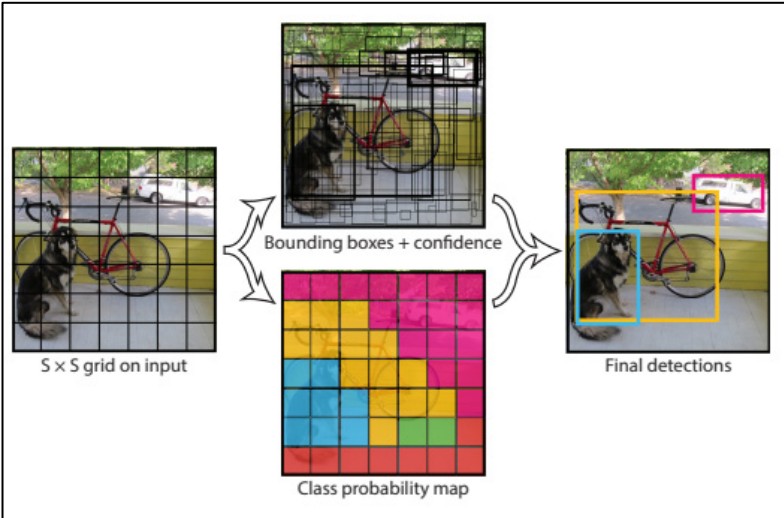

**Figure 2.** Bounding boxes and class probabilities for these boxes [14].

In its third version (v3—YOLO), the framework has a 106-layer neural network capable of detecting objects in three size scales, from small to very large, with nine anchor boxes

(three per size scale). This advancement has made YOLO particularly effective in detecting small objects in images [14]. Thus, YOLO represents an important milestone in the field of real-time object detection, combining speed and accuracy in object identification.

### 2.2. Fuzzy Sets

Classical logic, also known as Boolean logic, was developed by Aristotle and assigns only two states to its elements: belonging (membership degree 1) or not belonging (membership degree 0) to a specific set [21]. Unlike this dichotomous logic, fuzzy logic operates with degrees of truth varying between 0 and 1. For example, a membership degree of 0.5 may represent "half true" or "half false".

Unlike common mathematical variables that receive precise numerical values, in fuzzy logic, non-numerical values are employed to express a rule or a fact [22]. These values can be described as linguistic variables that take terms from natural languages, such as "few" or "many". Adjectives and adverbs are often used to expand the range of possible values, creating expressions like "very few people", "some people", "many people", and so on.

In fuzzy systems, input data go through a fuzzification process, which is the mapping of crisp values into a fuzzy space through membership functions. The output, in turn, goes through a defuzzification process, which converts the fuzzy value into a crisp value. This is particularly useful in control systems or decision-making, where an exact output value is needed [22].

### 2.3. Proposed Method

In this study, we use the YOLO framework for object detection in images. We chose YOLO for its efficiency, especially on low computational capacity devices, such as the Raspberry Pi, which was used in this project. Figure 3 illustrates the steps of the proposed system.

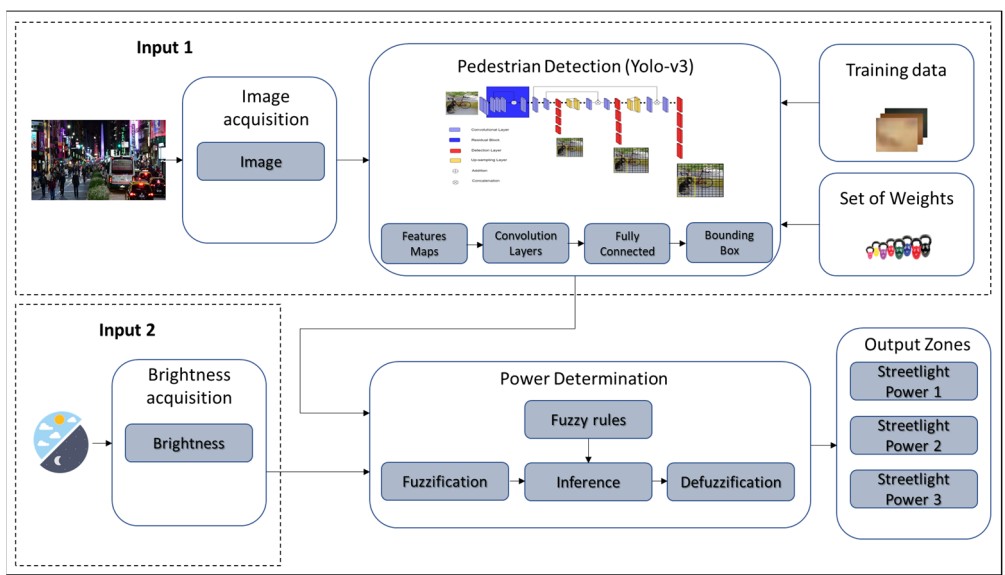

**Figure 3.** Details of the proposed method.

The system operates with two inputs to the fuzzy controller:

1. An image captured by a camera is pre-processed to adjust its dimensions to $608 \times 608$ pixels. Then, YOLO processes the image, identifying and counting the pedestrians detected in each region of the image;
2. The brightness index is recorded by a sensor and provided to the fuzzy controller.

Based on these two inputs, the fuzzy controller, following predefined inference rules, performs defuzzification, generating three outputs, one for each region of the image, to distribute the electrical power to each lighting unit.

The fuzzy controller sets the power following the minimum requirements of the NBR 5101:2018 [4] standard, ensuring adequate lighting of all roads, regardless of the number of pedestrians present.

Figure 4 illustrates the division of the three zones evaluated in the camera image, exemplifying how the power is adjusted at each lighting pole according to the number of pedestrians detected in these regions.

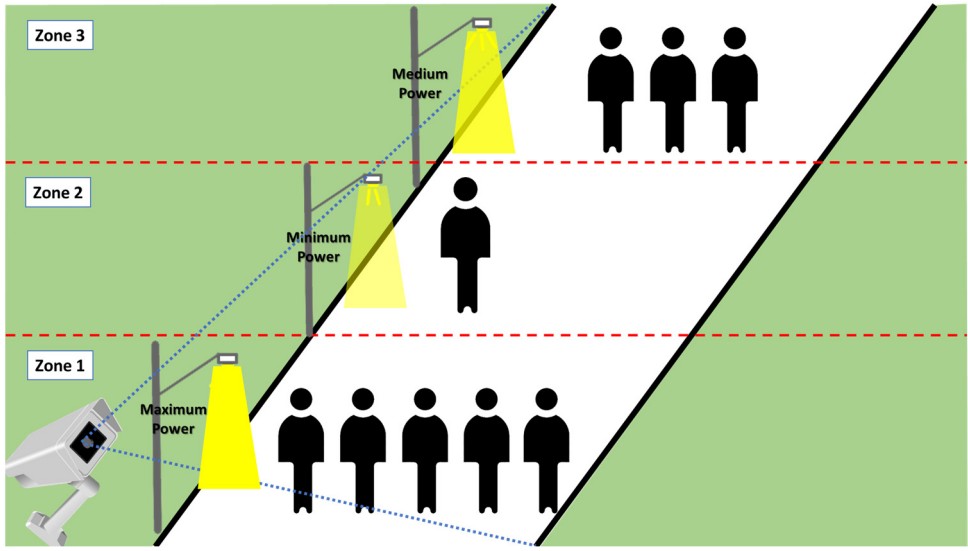

**Figure 4.** Delimitation of the 3 zones that will have the powers adjusted by the fuzzy controller.

In Figure 4, we also demonstrate the power output for each public lighting unit distributed in three equidistant zones of 30 m. This distribution reflects the minimum distance between the poles and adjusts the power based on the number of pedestrians in each region.

To meet the minimum operation parameters of the lighting units, we defined that the minimum power supplied to each unit is 30% of the total power, ensuring that the unit remains lit during the night, even in the absence of pedestrians.

Due to the ethical issue of people's anonymity, it is important to note that the images analyzed by the proposed system are not stored. After processing, they are discarded. This ensures that individuals detected will not be identified or tracked since there is no access to any database of individuals. Another pertinent detail regarding the security of the proposed system is that it should be under the use and control of public institutions. These institutions must ensure computer network security so that the information is not accessed improperly, ensuring no breach of privacy for anyone in the cameras' field of view.

However, it is worth noting that there are existing regulations about the use of cameras operated by public institutions in Brazil. For instance, in São Paulo, municipal law No. 17480 of 30 September 2020 [23] regulates the identification of individuals within the field of view of public cameras. These cameras can be used to identify infractions or crimes committed. Due to the complexity of the topic, the proposed system should be operated by public institutions or in enclosed areas with image rights and anonymity rules predefined.

### 2.4. Proposed Fuzzy System

In this fuzzy inference system, we define two input variables and one output variable. The input variables refer to the number of people on the road and the available natural light, while the output variable determines the lighting power to be provided by the luminaire. These three variables and their respective linguistic terms are presented in Figure 5.

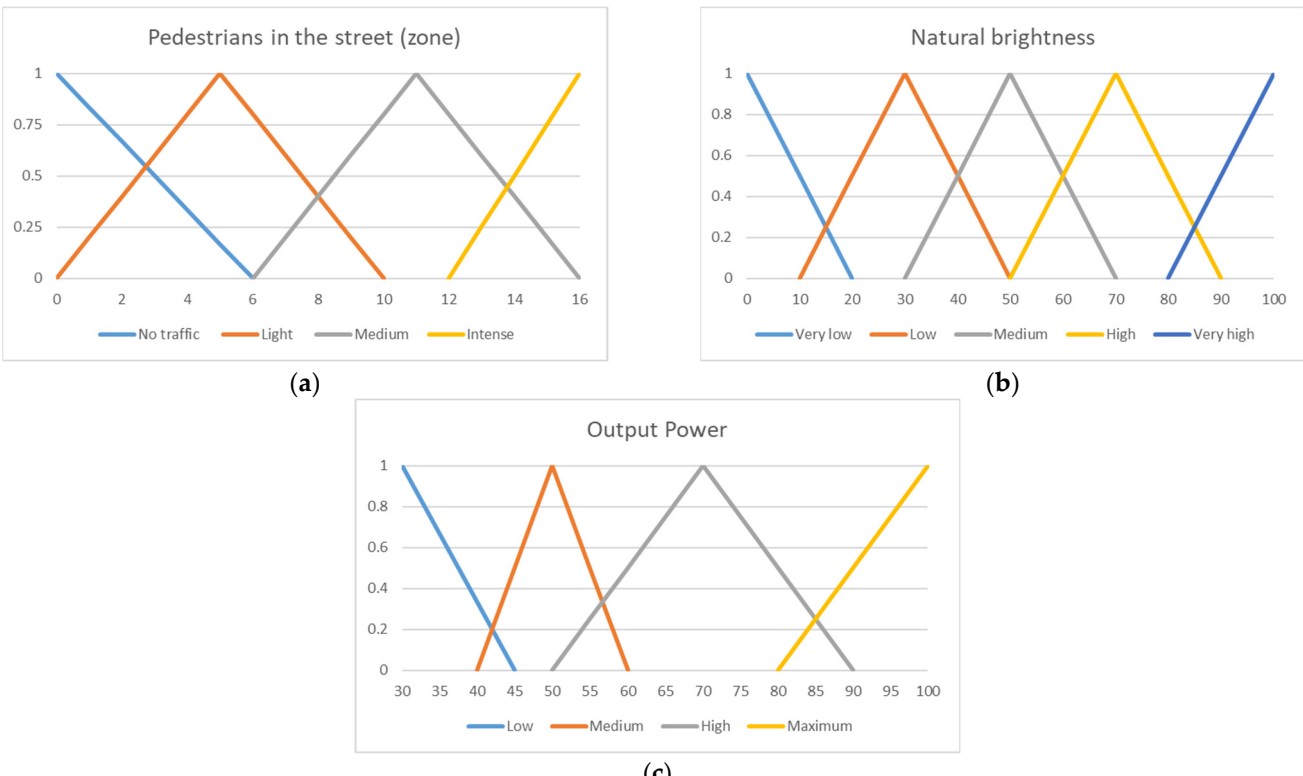

**Figure 5.** Membership Functions for the proposed fuzzy system. (**a**) Number of pedestrians in the street (zones); (**b**) natural brightness in percentage; and (**c**) output power in percentage.

The first input of the fuzzy system contains four linguistic terms represented by triangular membership functions, which indicate the number of people in the street: no traffic, light traffic, medium traffic, and intense traffic. These classifications are based on the groups of the NBR 5101:2018 [4] for pedestrian flow on roads, denominated P4, P3, P2, and P1, respectively. Considering that pedestrians move as they wish, not being forced to alter their movements due to the proximity of others, we adopt a volume of up to 16 pedestrians per minute per square meter on a 1.5 m wide sidewalk.

For the lighting existing in the environment, we created five linguistic terms (very low, low, medium, high, and very high) also represented by triangular membership functions. However, in this case, lighting is calculated as a percentage of natural brightness.

The output variable, shown in Figure 5c, represents the dimming of the artificial light power provided in percentage. This variable is also represented by four linguistic terms and triangular membership functions: low (P4), medium (P3), high (P2), and maximum (P1). Like the number of pedestrians, the power distribution is also based on the ABNT NBR 5101:2018 [4] standard. This ensures that, even in the absence of pedestrians, the lighting is at least 30%; that is, the lamps never completely turn off and always provide a minimum illuminance.

The powers were defined to correspond to the number of pedestrians present on the road. According to the NBR 5101:2018 [4] Standard, lighting levels are based on the width of pedestrian traffic lanes and not on the actual amount of these. Thus, the first level, denominated as low (P4), is intended for a very low number of pedestrians, maintaining minimum lighting as per the standard (3 LUX). As more pedestrian detections are processed, the subsequent levels are generated, providing an adequate level of lighting in the presence of pedestrians, and adjusting to the type of road classification and the minimum necessary lighting, from 3 to 20 LUX.

Table 3 shows the rules of the fuzzy system using the Takagi–Sugeno model [24]. The number of rules, which represent the output, is the product of the number of sets of each input variable, totaling 20 (5 sets of ambient lighting × 4 sets of the number of people present).

**Table 3.** Rules of the Fuzzy System.

| Natural Brightness/Pedestrian | No Traffic | Light | Medium | Intense |
|---|---|---|---|---|
| Very low | Low | Low | Low | Medium |
| Low | Low | Low | Medium | Medium |
| Medium | Low | Medium | Medium | High |
| High | Low | Medium | High | Maximum |
| Very high | Low | High | Maximum | Maximum |

*2.5. Simulation Scenario*

For the normal operation of the lamps, which are usually on for 12 h a day at maximum power, we will divide this period into four 3 h scenarios each. Each scenario corresponds to a range of natural luminosity (brightness): 0%, 30%, 70%, and 100%, where 0% represents total darkness and 100% total clarity. Each brightness range will represent a distinct test scenario.

The study targets implementation on secondary roads, where the movement of pedestrians per minute varies between 0 and 16 on a sidewalk up to 1.5 m wide. Power calculations will take into account LED lamps with a nominal power of 280 W.

Knowing the power of the lamp to be used, it is possible to calculate the consumption of each lighting unit in MWh by Equation (1), where $C$ is the consumption in MWh, $P$ is the nominal power in W, and $d$ is the number of days of operation. Thus, the individual consumption of each unit will be 0.00336 MWh per day.

$$C = \frac{P \times h \times d}{1,000,000} \tag{1}$$

With the consumption calculation in hand, we can price the cost of the public lighting system. Public lighting is classified in Group B (subgroup B4), defined in Clause XXXVIII of Art. 2 of Normative Resolution Aneel 414/2010.

Group B (low voltage) is composed of consumer units with supply at a voltage lower than 2.3 kV, charged by the monomial tariff (single rate of electricity consumption, regardless of the hours of use in the day). The tariffs applied to public lighting belong to subgroup B4 and are divided into tariffs $B_{4a}$ and $B_{4b}$.

The $B_{4a}$ tariff is applied when the public lighting assets, as well as the operation and maintenance services, are the responsibility of the municipality. This tariff represents only the energy consumption of the public lighting system. According to the distributor AES Eletropaulo, the cost per MWh of a $B_{4a}$ distribution network is USD 26.13/MWh. Therefore, we have (Equation (2)):

$$Cd = C \times B_{4a} \tag{2}$$

where $Cd$ is the cost per day of operation in USD, $B_{4a}$ is the tariff applied in a $B_{4a}$ type distribution network per MWh. Thus, we have that the cost per day of operation at maximum power of a unit is BRL 0.0948 per day or USD 2.81 per month.

As the operation period considered will be 12 h daily and there will be 4 daily scenarios in this period, each scenario will be applied at 3 h intervals. Therefore, the cost of R$ 0.094 per day will be divided into 4 parts, resulting in BRL 0.0235 for each 3 h period.

## 3. Results and Discussion

Our research allowed us to identify various techniques employed to deal with the same problem as this work; however, many of the systems used still leave pending issues. For instance, the management of public electric power consumption for lighting through

computer vision techniques is still a challenge, as previous studies, such as those by [13,25], restricted the application of these techniques to limited areas and private institutions.

The works of [16,26] address the issue of energy waste of street lighting in the absence of pedestrians. In particular, ref. [16] proposes an interesting solution, using the presence of pedestrians and the dimming of the power of the lamps. However, in his work, the posts furthest from the detected pedestrian are completely turned off. This may be inappropriate in a public situation, where it is preferable to maintain a minimum of lighting, regardless of the presence of pedestrians, to increase the feeling of safety.

In this sense, this work aims to fill gaps identified in previous studies by considering pedestrian density on a street. This information is essential for adapting the system to demand and for adjusting the power of the lamps according to existing technical standards on street lighting, aiming simultaneously for energy saving and adequate lighting.

### 3.1. Pedestrian Detection

For the conduct of the experiments, 2000 images of pedestrians were obtained through monitoring cameras installed in the city of Praia Grande, SP, Brazil. All images are public domain and do not allow the visual identification of the people depicted in them. The image bank was composed considering cameras in different positions and natural lighting conditions, with the images divided into three distinct lighting condition classes: day (100% natural lighting), dawn/dusk (transitional condition between light and dark), and night (absence of natural lighting).

Weather conditions were also taken into account. The image base was formed always considering favorable weather conditions for the acquisition of images and luminosity measurements; that is, clear days with little cloudiness.

When we used a single class called "People", the system presented a significant number of False Positives (FP)—1058 detections—indicating that the pedestrians were not correctly identified in the image. The Intersection over Union (IoU) rate shows that the system identified objects, but only 57.78% of these were actually considered pedestrians.

To evaluate which training model behaved better for the problem studied, we evaluated the reliability rate in the detection of some images from the base. The trained CNN presented an average reliability rate of 0.742, while the pre-trained network presented an average rate of 0.966.

Given these results, we chose to use a set of pre-trained weights provided by the framework itself, which includes various classes, such as People, Dogs, Cats, Bicycles, Cars, and Buses, among others, in order to evaluate the performance of our training in relation to the provided model.

The results show that the pre-trained model presented high-reliability rates, substantially superior to the trained model, even when the detected people were at a greater distance. Figure 6 shows an example of images processed with the three distinct lighting conditions.

Table 4 presents the actual number of pedestrians, the quantity detected, and the percentage of hits in the images. It should be noted that, for the validation of the experiments, we used different angles of image capture, thus enabling us to verify the reliability of the detections. The average reliability index for the detections, as reported by the algorithm, was 86.7%, which indicates good reliability in the detection of people. This index is calculated based on the average of each individual detected by the algorithm.

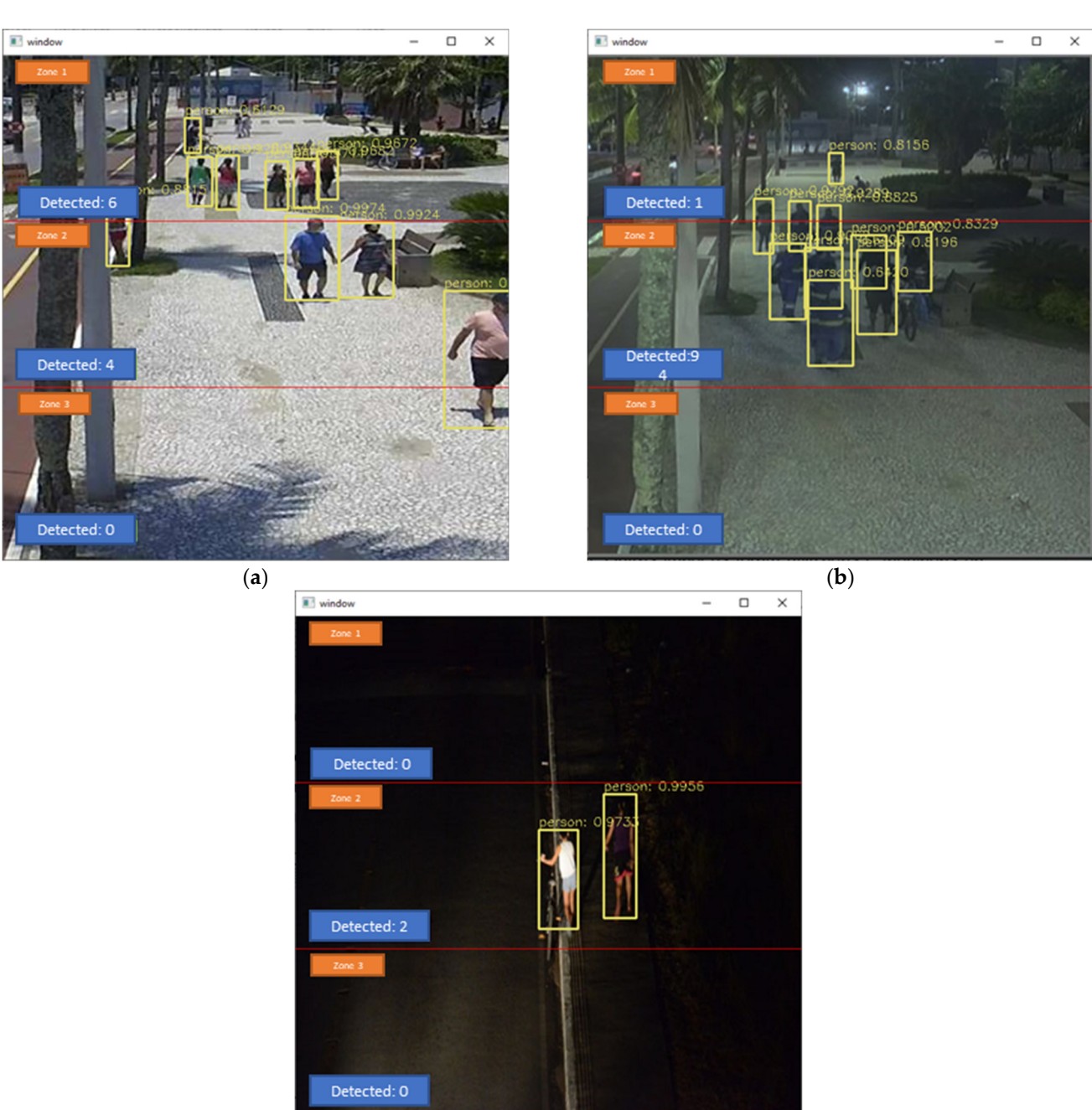

**Figure 6.** Examples of images processed. (**a**) Example with 100% of natural lighting; (**b**) example with 50% of natural lighting; and (**c**) example with 0% of natural lighting.

It is important to note that the accuracy rates fall under conditions of low natural light, as the images were captured with low brightness, often only with artificial lighting, and below the minimum proposed by the system, thus validating the minimum possible operation limits.

*3.2. Scenario Simulation*

Four scenarios were considered for the experiments, based on the amount of natural light during a 12 h operational period: 0%, 30%, 70%, and 100% natural lighting. The simulation scenarios were divided in this way to represent different levels of natural lighting and to obtain power outputs for the lighting set. It is important to note that the

calculations of system operating costs did not consider system maintenance costs, energy fluctuations, or equipment theft.

**Table 4.** Detection results of experiments with three natural lighting conditions.

| Image | 100% of Natural Lighting | | | 50% of Natural Lighting | | | 0% of Natural Lighting | | |
|---|---|---|---|---|---|---|---|---|---|
| | Pedestrians | Detection | Hits (%) | Pedestrians | Detection | Hits (%) | Pedestrians | Detection | Hits (%) |
| 1 | 20 | 18 | 90.0 | 2 | 2 | 100.0 | 18 | 13 | 72.2 |
| 2 | 15 | 10 | 66.7 | 10 | 9 | 90.0 | 17 | 7 | 41.2 |
| 3 | 12 | 10 | 83.3 | 2 | 2 | 100.0 | 17 | 10 | 58.8 |
| 4 | 7 | 5 | 71.4 | 8 | 6 | 75.0 | 19 | 9 | 47.4 |
| 5 | 7 | 5 | 71.4 | 10 | 9 | 90.0 | 16 | 6 | 37.5 |
| 6 | 11 | 8 | 72.7 | 10 | 8 | 80.0 | 8 | 5 | 62.5 |
| 7 | 13 | 11 | 84.6 | 6 | 6 | 100.0 | 8 | 3 | 37.5 |
| 8 | 12 | 10 | 83.3 | 22 | 18 | 81.8 | 13 | 8 | 61.5 |
| 9 | 12 | 10 | 83.3 | 8 | 7 | 87.5 | 18 | 10 | 55.6 |
| 10 | 8 | 8 | 100.0 | 6 | 5 | 83.3 | 17 | 5 | 29.4 |
| 11 | 5 | 4 | 80.0 | 23 | 20 | 87.0 | 5 | 5 | 100.0 |
| 12 | 7 | 6 | 85.7 | 12 | 8 | 66.7 | 5 | 4 | 80.0 |
| 13 | 14 | 13 | 92.9 | 10 | 9 | 90.0 | 5 | 3 | 60.0 |
| 14 | 22 | 21 | 95.5 | 7 | 7 | 100.0 | 10 | 4 | 40.0 |
| 15 | 34 | 29 | 85.3 | 11 | 9 | 81.8 | 5 | 3 | 60.0 |
| Total | 199 | 168 | 84.4 | 147 | 125 | 85.0 | 181 | 95 | 52.5 |

The presented values of the number of pedestrians per region represent a weekly average observed on a road where the validation images were captured. This value can and probably will vary according to the movement of the place. This value was used to illustrate the four scenarios to facilitate understanding.

- Scenario 1–0% natural lighting.

For each presented zone and number of pedestrians, the calculation was performed through the fuzzy inference system to obtain the lighting powers that will be necessary to meet the number of pedestrians detected in each region, as shown in Table 5.

**Table 5.** Scenario 1 results.

| Zone | Pedestrians qty | Simulated Power | Real Cost (USD) | Simulation Cost (USD) |
|---|---|---|---|---|
| 1 | 10 | 93.11% | 0.2355 | 0.02184 |
| 2 | 8 | 76.74% | 0.2355 | 0.01797 |
| 3 | 3 | 61.43% | 0.2355 | 0.01441 |
| Total | 21 | 77.09% | 0.0707 | 0.05424 |

- Scenario 2–30% natural lighting

In this scenario, we have the same number of pedestrians in each of the zones, and it is noticed that the system, when detecting that the natural lighting index is at 30%, adjusts the power variation to suit this condition. With the adjusted power variation, it is possible to verify that the simulated operation cost is significantly influenced, with the power reduced and suitable to the number of pedestrians on the road, as tabulated in Table 6.

**Table 6.** Scenario 2 results.

| Zone | Pedestrians qty | Simulated Power | Real Cost (USD) | Simulation Cost (USD) |
|---|---|---|---|---|
| 1 | 10 | 70.00% | 0.0236 | 0.01642 |
| 2 | 8 | 64.07% | 0.0236 | 0.01503 |
| 3 | 3 | 44.39% | 0.0236 | 0.01041 |
| Total | 21 | 59.49% | 0.0707 | 0.04185 |

- Scenario 3–70% natural lighting.

In this scenario, we have a condition very similar to dawn after 6 a.m., when the natural lighting index begins to increase rapidly. It is observed that the system continues to effectively calculate the power to be supplied based on these system inputs and the adjusted powers. With the increase of the lighting index, the power to be supplied to each region does not need to be so high, so the system reduces the power while pedestrians are detected. In Table 7, it is observed that the simulated cost drops approximately by half.

**Table 7.** Scenario 3 results.

| Zone | Pedestrians qty | Simulated Power | Real Cost (USD) | Simulation Cost (USD) |
|---|---|---|---|---|
| 1 | 10 | 50.00% | 0.0236 | 0.01173 |
| 2 | 8 | 50.00% | 0.0236 | 0.01173 |
| 3 | 3 | 44.39% | 0.0236 | 0.01041 |
| Total | 21 | 48.13% | 0.0707 | 0.03386 |

- Scenario 4–100% natural lighting

At this point, we have a good index of natural lighting, practically clear day, so the amount of power to be supplied should be low. The system calculates these indices, and it is noticed that the greatest power reduction occurs when the number of pedestrians is lower, and the natural lighting is high, but guaranteeing the minimum power required by the ABNT NBR 5101:2018 [4] standard, which is 30%.

Therefore, considering a scenario of 100% natural lighting, that is, almost completely clear, the system assigns a minimum power to all regions. At this point of natural lighting, the lighting system is about to turn off, coming to the end of the night operation period. Table 8 shows a significant reduction of this power, staying around 35% of the necessary.

**Table 8.** Scenario 4 results.

| Zone | Pedestrians qty | Simulated Power | Real Cost (USD) | Simulation Cost (USD) |
|---|---|---|---|---|
| 1 | 10 | 35.17% | 0.0236 | 0.00825 |
| 2 | 8 | 36.13% | 0.0236 | 0.00847 |
| 3 | 3 | 35.57% | 0.0236 | 0.00834 |
| Total | 21 | 35.62% | 0.0707 | 0.02506 |

After conducting the experiments, a consolidation of the obtained results was carried out, showing that the average power obtained with the system was 55.08%, the real operation cost for 12 h for three lighting units in a single day was USD 0.283, and the cost with the proposed system would be USD 0.1550 for the simulated values.

The daily cost for every three units of public lighting with the use of the proposed system shows that, the less natural lighting the environment receives, the cost tends to be slightly lower than the real (100% power). However, as the natural lighting index increases, the savings in cost increase, as it is not so necessary to provide power to the lighting units, thus saving electrical energy.

## 4. Conclusions

This research employed the YOLO framework and a fuzzy logic system, effectively optimizing public lighting management through computer vision. Despite initial challenges with the modest performance from individualized training, using a pre-trained weight set proved more successful, leading to enhanced pedestrian detection and energy efficiency.

A key contribution of this work is the incorporation of pedestrian density as a determinant for adaptive public lighting. Not only does this address gaps in previous research, but it also ensures compliance with technical lighting standards, marrying energy conservation with public safety.

Through simulated tests, our approach yielded an impressive 55.08% reduction in energy consumption for a 12 h operating window, compared to conventional methods. This significant achievement underscores the potential of our methodology, promising not just energy savings, but also improved pedestrian safety.

In closing, this study paves the way for further in-depth investigations into advanced, AI-driven solutions for sustainable and efficient public lighting, promising to redefine urban landscapes and contribute to the growth of smart cities.

**Author Contributions:** Conceptualization, A.S.V. and P.B.; methodology, A.S.V. and P.B.; software, A.S.V.; validation, A.S.V.; formal analysis, P.B.; investigation, A.S.V. and P.B.; resources, A.S.V.; data curation, A.S.V.; writing—original draft preparation, A.S.V. and P.B.; writing—review and editing, P.B.; visualization, A.S.V.; supervision, P.B.; project administration, P.B. All authors have read and agreed to the published version of the manuscript.

**Funding:** This research received no external funding.

**Data Availability Statement:** No new data was created or analyzed in this study. Data sharing does not apply to this article.

**Acknowledgments:** The authors would like to thank Nove de Julho University for financial support and the scholarship provided to Anderson Silva Vanin.

**Conflicts of Interest:** The authors declare no conflict of interest.

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
