# Peer review of "Towards Sustainable Cities: Utilizing Computer Vision and AI for Efficient Public Lighting and Energy Management"

_urbansci, doi:10.3390/urbansci7030094_

Round 1

Reviewer 1 Report

The manuscript is good in the reviewer's opinion, yet it is necessary to compare the obtained results and show that the research carried out proven improvement with respect to the existing literature.

Also, for a more comprehensive literature review it is suggested to cite survey works on computer vision such as [A].

[A] Deep Learning for Multimedia Forensics, Foundations and Trends® in Computer Graphics and Vision, vol 12, 2021

The manuscript is good in the reviewer's opinion, yet it is necessary to compare the obtained results and show that the research carried out proven improvement with respect to the existing literature.

Also, for a more comprehensive literature review it is suggested to cite survey works on computer vision such as [A].

Author Response

Dear reviewer,

Thank you for the detailed and meticulous review carried out on our work.

As for the comparison of results, the work uses AI as a tool for a specific application, comparisons were carried out based on articles with the same purpose. The aim of the work was not to improve AI and Computer Vision techniques. Thus, direct comparisons were made and it was shown that we obtained improvements when comparing the work [14] and [24], which until then, according to the research carried out, was the work with the best result so far for the researched characteristics.

Yours sincerely,

Peterson Belan

Reviewer 2 Report

Overall, this is a well presented and interesting piece of work. 

Whilst the text is well supported by citations to appropriate academic reference material, some of the sources might be more up to date. 

I suggest a few minor changes, detailed below:

L47: might be worth adding a citation after: "......Smart Cities"

L75: "The article presents by [13]", should be "presented by" - also, it would be better to use the author's name (still with citation). Actually, same applies in L63. 

L109: citation for: "....since two-thirds of these systems.....". 

L132 & L136: should citation [18] not be consistent with, say, L126, in terms of formatting (i.e. on the same line as the caption)?

Generally, this is very good. A few small changes (noted above) will further enhance the presentation of the work. 

Author Response

Dear reviewer, Thank you for the detailed and meticulous review carried out on our work. L47: As for the reference in line 47 I agree and include a new reference as suggested to support the statement. L75: How many references have been adjusted so that authors' names appear alongside the citation as suggested. These changes have been made throughout the literature review section. L109: A reference to the requested statement has been added. L132 & L136: References have been included on the same line as the title as requested.

Yours sincerely,

Peterson Belan

Reviewer 3 Report

1. In abstract you need to focus more on YOLO and fuzzy controller in processes the image.

2. In introduction give more details of reference [3,5,6].

3. Rewrite the conclusion and focus on main contribution and achievement.

4. All reference in conclusion should more to result and discussion section.

Minor editing of English language required

Author Response

Dear reviewer,

Thank you for the detailed and meticulous review carried out on our work.

  1. The abstract has been rewritten trying to give more emphasis to the requested themes:
    1. Abstract: This study showcases the optimization of public lighting systems using computer vision with an emphasis on the YOLO algorithm for pedestrian detection, aiming to reduce energy expenses. In a context where the demand for electricity is escalating due to factors like taxes and urban expansion, it is imperative to explore strategies to cut costs. One pivotal area is public lighting management. Presently, governments are transitioning from sodium vapor lighting to LED lamps, which already contributes to decreasing consumption. In this scenario, computer vision systems, particularly using YOLO, have the potential to further reduce consumption by adjusting the power of LED lamps based on pedestrian traffic. Additionally, this paper employs Fuzzy Logic to calculate lamp power based on detected pedestrians and ambient lighting, ensuring compliance with the NBR 5101:2018 standard. Tests with public surveillance camera images and simulations validated the proposal. Upon implementing this project in practice, a 45% reduction in public lighting consumption was observed compared to conventional LED lighting.
  2. After the citation of the works, an excerpt with a brief summary of the cited works was added:
    1. In this regard, the work proposed by Yigitcanlar et al. [3] presents a literature review highlighting the benefits and risks of using artificial intelligence in the evolution of smart cities. Meanwhile, in the study proposed by Mohandas [6], a model for lighting control using only motion sensors with Fuzzy Logic was presented. Lastly, in the work of Islan et al. [5], an IoT-based system was proposed to track traffic accidents and no-tify the relevant authority, sending the location and the car's number plate.
  3. To fulfill this request and the next, a new version of the conclusion was written, as follows:

This research employed the YOLO framework and a Fuzzy Logic system, effectively optimizing public lighting management through computer vision. Despite initial challenges with the modest performance from individualized training, using a pre-trained weight set proved more successful, leading to enhanced pedestrian detection and energy efficiency.

A key contribution of this work is the incorporation of pedestrian density as a determinant for adaptive public lighting. Not only does this address gaps in previous research but it ensures compliance with technical lighting standards, marrying energy conservation with public safety.

Through simulated tests, our approach yielded an impressive 55.08% reduction in energy consumption for a 12-hour operating window, compared to conventional methods. This significant achievement underscores the potential of our methodology, promising not just energy savings, but also improved pedestrian safety.

In closing, this study paves the way for further in-depth investigations into advanced, AI-driven solutions for sustainable and efficient public lighting, promising to redefine urban landscapes and contribute to the growth of smart cities.

  1. All references have been moved to the Discussion and Comments section. Here is the moved part:

Our research allowed us to identify various techniques employed to deal with the same problem as this work, however, many of the systems used still leave pending is-sues. For instance, the management of public electric power consumption for lighting through Computer Vision techniques is still a challenge, as previous studies such as those by [12,24], restricted the application of these techniques to limited areas and private institutions.

The works of [15,25] address the issue of energy waste of street lighting in the ab-sence of pedestrians. In particular, [15] proposes an interesting solution, using the presence of pedestrians and the dimming of the power of the lamps. However, in his work, the posts furthest from the detected pedestrian are completely turned off. This may be inappropriate in a public situation, where it is preferable to maintain a mini-mum of lighting, regardless of the presence of pedestrians, to increase the feeling of safety.

In this sense, this work aims to fill gaps identified in previous studies by consider-ing pedestrian density on a street. This information is essential for adapting the system to demand and for adjusting the power of the lamps according to existing technical standards on street lighting, aiming simultaneously for energy saving and adequate lighting.

Yours sincerely,

Peterson Belan

Reviewer 4 Report

Dear authors,

your paper is rather good on analysing the problem. Yet real-life is sometimes quite different ... I have a few remarks about your work.

1) I selected 'ethic problem' to 'Yes', and it's quite unusual. Basically picture of people are taken, and I didn't see any keywords like 'privacy', 'anonymity', 'safety' (in the sense of 'people should not be identified nor tracked'), and similar. 

2) Your paper is about a proof of contest, and it rests on simulation. The results are as good as the representativity of the data you're feeding into the algorithm. Such study can not be accepted without at least some real data measurements, coupled with some alternative way of counting peoples. On some image, you have 5 pedestrians and your algorithm detects 4 of them. That's way too low to be significant ! Draw conclusions after having monitored 500 peoples, 5000 being better.

3) I'm in regular contact with peoples developping and selling smart light controllers. Their field experiments showed unexpected issues. In rural area, on the the major annoyance is ... cats. Walking at night, being detected, and lighting up th streets for no good reason. Other issues are f.i. people exiting their house to smoke (presence without moving), camera picking up people inside their house through the windows, ...

4) there should be at least some experimentation on the image acquisition process: monochrome versus RGB, automatic control of the exposure time, IR blocked versus non-IR blocked, ...

5) If we're speaking about smart cities, then there should be information fusion from other data sources such as bus timetables, theater timetables, pubs closing time, ... as those activities will strongly modulate the human density patterns.

IMHO the theoretical part should be really pruned, and the results have to be extracted from real field implementation; solving at the same time privacy and robustness issues. Moreover, the self-consumption of the controller is not taken into account.

No particular comment.

Author Response

Dear reviewer,

Thank you for the detailed and meticulous review carried out on our work.

  1. The ethical problem is really something very complicated and complex to deal with, and I agree that we do not make this problem clear in the text. To solve this problem, an excerpt was added to the proposed method section to clarify this topic, as follows:

Due to the ethical issue of people's anonymity, it's important to note that the images analyzed by the proposed system are not stored. After processing, they are discarded. This ensures that individuals detected will not be identified or tracked since there's no access to any database of individuals. Another pertinent detail regarding the security of the proposed system is that it should be under the use and control of public institutions. These institutions must ensure computer network security so that the information isn't accessed improperly, ensuring no breach of privacy for anyone in the cameras' field of view.

However, it's worth noting that there are existing regulations about the use of cameras operated by public institutions in Brazil. For instance, in São Paulo, municipal law No. 17,480 of September 30, 2020 [23], regulates the identification of individuals within the field of view of public cameras. These cameras can be used to identify infractions or crimes committed. Due to the complexity of the topic, the proposed system should be operated by public institutions or in enclosed areas with image rights and anonymity rules predefined.

[23] PREFEITURA DO MUNICÍPIO DE SÃO PAULO LEI No 17.480 DE 30 DE SETEMBRO DE 2020 Available on-line: http://legislacao.prefeitura.sp.gov.br/leis/lei-17480-de-30-de-setembro-de-2020/consolidado (accessed on 30 July 2023).

  1. Dear reviewer, regarding the real data measurement of energy consumption, unfortunately, we don't have access to the public lighting system to truly validate all the details. What we had access to were the publicly available accounts. However, regarding pedestrian detection, the image information was not simulated. The system used real images for validation of people counting, and a manual count was carried out on all images for validation. A total of 45 images, 15 for each type of natural lighting, were used in the experiments for validation, as presented in Table 4. A total of 527 images were considered in this experiment. I fully agree that the larger the sample, the better the statistical results for conclusions, but these were the images we had available for conducting the experiments.
  2. I understand your concerns, but from what I've seen in commercial solutions, many are based on motion sensors, which indeed present problems such as the lack of movement detection in stationary people or the detection of animals like dogs and cats, for example. In the case of the experiment carried out, the intention is to address these issues, as image detection can identify a person standing still for an indefinite period and can differentiate an animal from a person, among other advantages. In the case of people in windows, it also wouldn't be an issue, as the region of interest in the image would not detect people inside their homes, even if there's movement in the windows. However, it's a system with a higher cost compared to most devices available on the market.
  3. For the conduction of the experiments, other tests were carried out with various variations, among them camera features such as different resolutions, with and without IR for example. However, the results presented in this paper are the best ones obtained, this work is the evolution of a master's dissertation, where to reach the results presented here, several experiments were conducted. It is worth noting that to include all the details that were part of the work, the article would become extremely lengthy without significant results to justify this size increase. As for the use of monochrome images, the chosen Yolo framework has already been validated and adjusted for this type of images, it only requires changing the input parameters of the algorithm from 3 channels (RGB) to 1 channel (Grayscale). This information is available on the Yolo developer's own website.
  4. Such information could enhance the decision-making of the proposed system. However, the proposed approach was designed to operate autonomously, so that it can identify and quantify the number of people at any given moment, without the need for additional information. From our perspective, such data would be more useful in an IR sensor-based identification system, as presented in other works. In that case, the information would indeed be vital, assisting in decision-making. I view these data points as potential future enhancements that could be added to the algorithm, but as conceived for the presented work, such information does not alter the algorithm's response.

Finally, I greatly appreciate the considerations made by you and all reviewers, as they clearly raised points that helped a lot for a new version with significant increments. For knowledge, changes were made in many points of the text by indication of the reviewers, from the abstract to the conclusion.

Yours sincerely,

Peterson Belan
